# Late Changes in Renal Volume and Function after Proton Beam Therapy in Pediatric and Adult Patients: Children Show Significant Renal Atrophy but Deterioration of Renal Function Is Minimal in the Long-Term in Both Groups

**DOI:** 10.3390/cancers16091634

**Published:** 2024-04-24

**Authors:** Yinuo Li, Masashi Mizumoto, Hazuki Nitta, Hiroko Fukushima, Ryoko Suzuki, Sho Hosaka, Yuni Yamaki, Motohiro Murakami, Keiichiro Baba, Masatoshi Nakamura, Toshiki Ishida, Hirokazu Makishima, Takashi Iizumi, Takashi Saito, Haruko Numajiri, Kei Nakai, Satoshi Kamizawa, Chie Kawano, Yoshiko Oshiro, Hideyuki Sakurai

**Affiliations:** 1Department of Radiation Oncology, University of Tsukuba, Tsukuba 305-8577, Ibaraki, Japan; yli@pmrc.tsukuba.ac.jp (Y.L.); hnitta@pmrc.tsukuba.ac.jp (H.N.); murakami@pmrc.tsukuba.ac.jp (M.M.); nakamura@pmrc.tsukuba.ac.jp (M.N.); tishida@pmrc.tsukuba.ac.jp (T.I.); hmakishima@pmrc.tsukuba.ac.jp (H.M.); iizumi@pmrc.tsukuba.ac.jp (T.I.); saitoh@pmrc.tsukuba.ac.jp (T.S.); haruko@pmrc.tsukuba.ac.jp (H.N.); knakai@pmrc.tsukuba.ac.jp (K.N.); kamizawa@pmrc.tsukuba.ac.jp (S.K.); kawano@pmrc.tsukuba.ac.jp (C.K.); hsakurai@pmrc.tsukuba.ac.jp (H.S.); 2Department of Child Health, Institute of Medicine, University of Tsukuba, Tsukuba 305-8577, Ibaraki, Japan; fkhiroko@md.tsukuba.ac.jp (H.F.); ryokosuzuki@md.tsukuba.ac.jp (R.S.); 3Department of Pediatrics, University of Tsukuba Hospital, Tsukuba 305-8576, Ibaraki, Japan; shohosaka@md.tsukuba.ac.jp (S.H.); yamaki.yuni.ru@alumni.tsukuba.ac.jp (Y.Y.); 4Department of Radiation Oncology, Tsukuba Medical Center Hospital, Tsukuba 305-8558, Ibaraki, Japan; ooyoshiko@pmrc.tsukuba.ac.jp

**Keywords:** proton beam therapy, pediatric, adult, kidney volume, renal function, late effects

## Abstract

**Simple Summary:**

Understanding the age-specific effects of proton beam therapy (PBT) on kidney function is crucial for personalized treatment. This study compared the long-term impact of PBT on kidneys in pediatric and adult patients with adjacent malignancies. Findings reveal that children are more prone to renal atrophy post-PBT compared to adults, who experience minimal changes in kidney morphology. The percentage of irradiated volume receiving 10 Gy (RBE) and 20 Gy (RBE) may predict the degree of renal atrophy, especially in children. PBT has minimal impact on renal function deterioration in both age groups. This research informs treatment plans for patients in different ages, aiming to minimize renal complications and provide insights for personalized cancer care, ensuring better long-term kidney health outcomes and overall quality of life.

**Abstract:**

To compare late renal effects in pediatric and adult patients with malignancies after PBT involving part of the kidney. A retrospective study was conducted to assess changes in renal volume and function in 24 patients, including 12 children (1–14 years old) and 12 adults (51–80 years old). Kidney volumes were measured from CT or MRI images during follow-up. Dose-volume histograms were calculated using a treatment planning system. In children, the median volume changes for the irradiated and control kidneys were −5.58 (−94.95 to +4.79) and +14.92 (−19.45 to +53.89) mL, respectively, with a relative volume change of −28.38 (−119.45 to −3.87) mL for the irradiated kidneys. For adults, these volume changes were −22.43 (−68.7 to −3.48) and −21.56 (−57.26 to −0.16) mL, respectively, with a relative volume change of −5.83 (−28.85 to +30.92) mL. Control kidneys in children exhibited a marked increase in size, while those in adults showed slight volumetric loss. The percentage of irradiated volume receiving 10 Gy (RBE) (V10) and 20 Gy (RBE) (V20) were significantly negatively associated with the relative volume change per year, especially in children. The CKD stage based on eGFR for all patients ranged from 1 to 3 and no cases with severe renal dysfunction were found before or after PBT. Late effects on the kidneys after PBT vary among age groups. Children are more susceptible than adults to significant renal atrophy after PBT. V10 and V20 might serve as predictors of the degree of renal atrophy after PBT, especially in children. PBT has a minimal impact on deterioration of renal function in both children and adults.

## 1. Introduction

Patients with cancer commonly receive multidisciplinary treatment, including surgery, radiotherapy, and chemotherapy [1]. Radiotherapy is crucial for local control of tumors; however, it may also result in long-term adverse events such as radionecrosis, fistulas, and secondary malignancies [2,3]. Proton beam therapy (PBT) is a progressive iteration of radiotherapy that, due to its excellent Bragg peak dose distribution, allows minimization or elimination of radiation doses to normal tissues close to the tumor target while delivering a high dose to the tumor [4]. Thus, PBT can lead to less severe side effects through minimizing short- and long-term damage to the intestines, kidneys, bladder, rectum, bone marrow [5,6,7].PBT has also been shown to be superior in multidisciplinary management of pediatric oncology [8] and may diminish the risk of growth retardation, endocrine disorders, decreased fertility, and secondary cancers in children and adolescents and young adults (AYAs) [9]. However, PBT also resulted in late effects in some tissues, even at low irradiation doses; for example, vertebrae in children receiving PBT showed impaired bone growth, with a significantly reduced growth rate compared to non-irradiated vertebrae [10]. In a study of proton therapy in pediatric patients younger than 5 years of age, it was demonstrated that muscle growth tended to decrease with increasing irradiation dose in subsequent follow-up observations, with little muscle growth at doses greater than 50 Gy (RBE) [11].

The high sensitivity of the kidneys to radiation is widely recognized; indeed, the kidneys are the primary bystander organ limiting the dose in radiotherapy planning [12]. In PBT for abdominal tumors such as hepatocellular carcinoma (HCC) in adults, the kidneys are commonly exposed unilaterally or bilaterally in irradiation of the primary lesion [13]. Thus, radiation-associated kidney disease is a delayed consequence of abdominal radiotherapy and is frequently characterized by increased levels of serum creatinine, proteinuria, anemia, and hypertension [14]. Moreover, the total parenchymal volume of kidneys exposed to radiation consistently diminishes from pre-treatment to 36–94 months post-treatment, with size reduction of 23% (26 mm) and volume loss of 47% (91 mL) [15].

PBT is currently available for treatment of only a few types of cancers, compared to conventional modalities. As a highly conformal type of radiotherapy, PBT is used by a relatively small number of healthcare professionals in a limited number of cases. There have been several studies of the late effects of photon-based radiotherapy on the kidneys [16,17], but effects on the kidneys after PBT are not fully understood and more studies are needed, especially with regard to variability among age cohorts. Therefore, the aim of this study was to evaluate changes in renal volume and function in pediatric and adult patients undergoing PBT and to explore correlations with dosimetric parameters.

## 2. Materials and Methods

### 2.1. Patients

A total of 24 patients who underwent PBT between October 2010 and October 2020 were enrolled in the study, including 12 children (6 males, 6 females) and 12 adults (6 males, 6 females) with median ages at the start of PBT of 3.5 years (range: 1–14 years) and 66 years (range: 51–80 years), respectively. The primary diseases of the pediatric patients were neuroblastoma (*n* = 8), rhabdomyosarcoma (*n* = 2), osteosarcoma (*n* = 1), and Ewing sarcoma (*n* = 1). For adults, the primary diseases were HCC (*n* = 6), metastatic liver cancer (*n* = 4), and pancreatic cancer (*n* = 2). Nine of the children underwent gross total resection (GTR) prior to radiotherapy, one underwent subtotal resection (STR), and two had biopsies. Only one of the adults received STR. All the children received chemotherapy, including pre-radiation chemotherapy (*n* = 7) and concomitant chemoradiotherapy (*n* = 5). Of the adults, 5 received chemoradiotherapy and 1 had pre-radiation chemotherapy.

All procedures involving human participants in this study, encompassing case reviews, adhered to the ethical standards outlined in the 1964 Declaration of Helsinki and its subsequent amendments (or equivalent ethical standards). The study received approval from the Institutional Review Board at Tsukuba Clinical Research & Development Organization. The treatment plans for all cases underwent thorough deliberation at in-hospital conferences, and informed consent was obtained from all subjects (or parents) in the study.

### 2.2. Proton Beam Therapy

Pre-treatment computed tomography (CT) was used in treatment planning for all patients to provide a baseline for the initial organ morphology. The booster synchrotron at the University of Tsukuba generates 250 MeV of proton beam energy. The allocation of dose fractions was based on the tumor location and the presence of organs at risk. The treatment planning system delineated the dose distribution, collimator configuration, bolus, and range-shifter thickness settings. The relative biological effectiveness (RBE) was defined as 1.1 [18]. PBT was administered once a day on weekdays. Due to the location of the primary lesion, the irradiation field involved the unilateral kidneys in all cases. Pediatric and adult cases received doses of 19.8–70.4 Gy (RBE) (median: 30.6 Gy (RBE)) and 66–74 Gy (RBE) (median: 72.6 Gy (RBE)), respectively. The characteristics of the patients, primary diseases, and PBT are shown in Table 1.

### 2.3. Analysis of Renal Volume and DVH

Mid-to-long-term follow-up after PBT is required to assess the treatment efficacy and effects on organs. The median follow-up periods were 32.1 months (range: 23.5–56.5) for pediatric cases and 70.8 months (range 62.7–136.5) for adult cases. Renal morphology was observed in pre-PBT CT and CT or magnetic resonance imaging (MRI) performed at follow-up visits. MIM Maestro (ver. 6.5.2) software was used for kidney contouring and volume calculations. Dose-volume histograms (DVHs) for each patient were obtained from the treatment planning system using VQA ver. 2 software. Dosimetric parameter analysis encompassed V10 (% of irradiated volume receiving 10 Gy (RBE)), V20, and V30.

### 2.4. Kidney Function

The glomerular filtration rate (GFR) is a vital indicator for assessing kidney function. Chronic kidney disease (CKD) staging for adults is based on the KDIGO CKD guidelines 2012, modified for Japanese standards. For children older than 2 years of age, CKD staging is determined based on the absolute value of the estimated GFR (eGFR). Serum creatinine (s-Cr) and eGFR levels were recorded in blood tests obtained at our hospital or at a follow-up hospital. For a few pediatric patients without eGFR values, s-Cr-based extrapolation was applied to obtain eGFR [19]. Based on the Staging and Classification of Pediatric CKD in Japan (≥2 years old) published by The Japanese Society of Pediatric Nephrology, renal function was assessed before, during, and after PBT. For patients aged >3 months but younger than 2 years old with low physiologic GFR, %GFR was compared with the GFR reference value (median) at each month of age to determine the staging.

### 2.5. Statistical Analysis

Data were initially compiled using a Microsoft Excel 2010 spreadsheet. Statistical analysis was performed using SPSS ver. 26.0 for Windows. Pearson correlation analysis was used to investigate correlations and changes in the irradiated and control kidney volumes in follow-up after PBT, and correlations of the relative volume change per year with V10 and V20 in different age groups (adults and children). A larger absolute value of the Pearson coefficient (r) indicates a stronger correlation. Statistical significance was set at *p* < 0.05.

## 3. Results

### 3.1. Treatment Characteristics and Renal Volume Changes

The 24 patients with malignancies treated with PBT from 2010 to 2020 comprised 12 children (1–14 years old) and 12 adults (51–80 years old) with median follow-up periods of 32.1 (range: 23.5–56.5) and 70.8 (range 62.7–136.5) months, respectively. The irradiation doses were 19.8–70.4 Gy (RBE) (median: 30.6 Gy (RBE)) in 11–32 fractions (median: 17) for children and 66–74 Gy (RBE) (median: 72.6 Gy (RBE)) in 10–50 fractions (median: 22) for adults. In the pediatric group, 11 patients received a dose of 1.8 Gy (RBE) per fraction, and 1 patient received 2.2 Gy (RBE) per fraction. As for the irradiation paths, 8 children were treated with a single posterior-anterior field, 2 received two-field irradiation from anterior and posterior directions, and the remaining 2 children were irradiated by two-field protons from both the posterior–anterior and right oblique directions. For adults, the irradiation was 3.3 Gy (RBE) per fraction for 9 patients, 2 Gy (RBE) for 1 patient, and additionally, there were 2 pancreatic cancer patients whose treatment plans were changed from 2 Gy (RBE) per fraction to 0.7 Gy (RBE) per fraction with small field irradiation. The adult irradiation paths were two-fields from the direct right side and oblique posterior direction (*n* = 10), and two-fields from anterior and posterior directions (*n* = 2). Due to the adjacency of the primary lesion, the unilateral kidneys were involved in the irradiation field in all cases. The background disease, PBT doses, V10 to V30, and kidney volume changes are shown for each pediatric case in Table 2 and for each adult case in Table 3.

The initial renal volume at the start of PBT was used as a reference. As shown in Table 2, the median volume changes at the last follow-up in pediatric patients were −5.58 (range: −94.95 to +4.79) mL for irradiated kidneys and +14.92 (−19.45 to +53.89) mL for control kidneys. Over the follow-up period, the volume of irradiated kidneys decreased in 7 children by −31.08 (−94.95 to −5.24) mL and increased in 5 children by +3.52 (+0.67 to +4.79) mL. In these 5 children, the increase was not as great as that for the contralateral control kidneys, which increased by +14.4 (+5.96 to +53.89) mL. For the control kidneys, the volume increased in 10 children by +18.87 (+5.6 to +53.89) mL and decreased in 2 children by −11.74 (−19.45 to −4.03) mL; similarly, the decrease in these children was much less than that for the irradiated kidneys, which decreased by −34.13 (−49.51 to −18.74) mL. Therefore, using the control kidneys as the reference, the irradiated kidneys of all 12 pediatric patients showed a uniformly significant trend for atrophy after PBT. The relative volume change in the irradiated kidneys was −28.38 (−119.45 to −3.87) mL.

The median volume change at the last follow-up in adults was −22.43 (range: −68.7 to −3.48) mL for irradiated kidneys and −21.56 (−57.26 to −0.16) mL for control kidneys (Table 3). Over the follow-up period, the volume of irradiated kidneys decreased in all 12 adults, and all control kidneys were also reduced in size. The relative volume change in the irradiated kidneys was −5.83 (−28.85 to +30.92 mL), with an increase of +18.03 (+2.54 to +30.92) mL in 5 cases, and a decrease of −11.86 (−28.85 to −1.1) mL in 7 cases. Unlike the pediatric cases, there were no uniform and regular relative changes in the volume of the irradiated kidneys in the adult cases after PBT.

Pearson analysis of the relative volume changes in the irradiated kidneys in the pediatric and adult cases is shown in Figure 1a. The relative volume change in pediatric cases had a negative linear relationship with follow-up time, indicating significant and consistent atrophy of the irradiated kidneys. In contrast, the relative volume reduction (that is, atrophy) was not pronounced in adult cases. Thus, the responses of the irradiated kidneys after PBT clearly differed in pediatric and adult cases. The volume change in the control kidneys in the two age groups is shown in Figure 1b. The control kidneys gradually became larger as the children grew and matured, whereas those of adults showed very slight atrophy, but the trend for atrophy was relatively stable over time. Thus, as for the irradiated kidneys, there were clear differences in the responses of the control kidneys in pediatric and adult cases after PBT.

### 3.2. DVH Analysis

As shown in Table 2 and Table 3, most pediatric patients and all adults received PBT exposures of 10 Gy (RBE) and 20 Gy (RBE) (V10, V20). The Pearson correlation coefficients (r) and the significance (*p*) levels for relationships of V10 and V20 with relative volume changes in irradiated kidneys per year are summarized in Table 4. The relative volume change in the kidneys after irradiation in all patients showed the strongest significant linear correlation with V10 (r = −0.701, *p* < 0.001) and also showed correlation with V20 (r = −0.559, *p* = 0.005). Thus, as V10 or V20 becomes larger, the irradiated kidney loses more volume and atrophy is more severe. Correlation analyses of V10 and V20 with relative volume changes in pediatric and adult cases are shown in Figure 2. V10 showed a strong negative linear correlation with relative volume change in pediatric cases (r = −0.475, Figure 2a). Thus, the larger the V10, the more severe the atrophy of the irradiated kidneys in children. A weaker correlation was also present in adult cases (r = −0.312). V20 also had a strong linear correlation with relative volume change in pediatric cases (r = −0.443, Figure 2b), but only a weak correlation in adult cases (r = −0.121). None of these relationships were found to be statistically significant.

### 3.3. Renal Function

GFR is an important indicator of renal function. s-Cr and eGFR were recorded before, during, and after PBT (until about 12 months after completion of irradiation). For pediatric patients without documented eGFR data, eGFR was derived using an extrapolation method based on s-Cr [19]. Changes in the number of patients according to the eGFR-based CKD classification in different time periods are shown in Table 5. In accordance with the Staging and Classification of Pediatric CKD in Japan and KDIGO CKD guidelines 2012 modified for Japanese patients, the criteria for determining renal function and CKD staging were as follows: stage 1, GFR ≥ 90 mL/1.73 m^2^, indicating normal or hyperactive renal function; stage 2, GFR 60–89 mL/min/1.73 m^2^, normal or mildly depressed renal function; stage 3, GFR 30–59 mL/min/1.73 m^2^, moderately depressed renal function; stage 4, GFR 15–29 mL/min/1.73 m^2^, highly depressed renal function; stage 5, GFR < 15 mL/min/1.73 m^2^, end-stage kidney disease (ESKD). Before, during, and after irradiation, all adult patients were in stages 1 to 3, whereas pediatric patients were all in stages 1 to 2 (Table 5). At the follow-up visit after completion of PBT, only one more adult had worsened to stage 3. Thus, renal function remained unchanged in most patients. These results suggest that PBT has little effect on renal function in adults or children, and no patients had severe renal failure.

Prostate diseases in males, such as benign prostatic hyperplasia (BPH), can impact kidney function. Of the six adult males in this study, only one patient developed BPH during the post-PBT follow-up, which did not cause urinary obstruction but was accompanied by an elevated PSA level. It raised suspicion of prostate cancer but did not confirm the diagnosis. Three patients showed no symptoms of BPH, and two patients did not have prostate CT or MRI available, thus their prostate condition could not be assessed.

## 4. Discussion

Radiotherapy is a common cancer treatment modality, particularly for tumors that are localized or fully encompassed within the radiation field [20]. However, radiotherapy cannot selectively destroy cancer cells without affecting adjacent normal cells, potentially leading to long-term adverse events, including secondary cancers [21]. In head and neck cancer, late adverse events after radiotherapy include osteoradionecrosis, radiation recall myositis, oral necrosis, skin cancer, and thyroid dysfunction [22]. At the prevailing doses used for management of high-risk embryonal tumors of the central nervous system, craniospinal irradiation (CSI) can influence the height of pediatric patients [23]. Additionally, with high-dose irradiation, there is a noticeable reduction in vertebral growth, and even with symmetrical irradiation, there is also a high risk of developing spinal curvature after CSI [24]. Even in proton therapy, high-dose postoperative irradiation of chondrosarcoma of the skull base may lead to adverse effects such as temporal lobe necrosis, which has raised further concerns among clinicians regarding the evaluation of side effects after radiotherapy [25].

Kidney involvement in irradiation occurs when the primary tumor is in or close to the kidneys. Progression of renal atrophy and renal insufficiency has been noted in long-term follow-up after radiotherapy [26] and the degree of renal atrophy may be dose-distribution dependent [27]. Tran et al. found that adult kidneys subjected to irradiation during treatment of a neighboring abdominal malignancy showed persistent significant atrophy over a follow-up period of almost 8 years. The volume changes were twice as great as the length changes, leading to decreased renal function [15].

Radiation nephropathy refers to renal injury and functional impairment induced by ionizing radiation and may result in renal failure. Acute radiation nephropathy typically manifests from 6 months after radiotherapy, and presents with symptoms such as hypertension, azotemia, asymptomatic proteinuria, malaise, or anemia [28]. Chronic radiation nephropathy typically emerges around 18 months post-radiotherapy and is characterized by diminished kidney volume, hypertension, proteinuria, and anemia [29]. Doses below 18 Gy to the entire kidney seldom lead to severe or enduring renal damage, while doses exceeding 20 Gy result in notable nephropathy [30]. V5 in the irradiated kidneys has been suggested to have the strongest association with severe chronic renal failure of grade 2 or higher, based on a significantly higher 5-year incidence in patients with V5 ≥ 58%. This implies that V5 is particularly relevant to the risk of developing CKD [14]. In a post-SBRT study of pancreatic cancer, V5 of the irradiated kidney >210 cm^3^ predicted a >23 mL/min/1.73 m^2^ decrease in GFR after SBRT for pancreatic cancer, indicating that V5 is a predictor of significant nephrotoxicity after SBRT [31]. Additionally, in a long-term evaluation of renal function in children after photon radiotherapy, 126 patients (median age: 10.2 years) received radiotherapy to parts of the kidneys. The results indicated that in the current multimodal pediatric treatments, kidney impairment induced by radiotherapy is rare in children, but the occurrence of grade 1 toxicity is significantly related to the irradiated volume of the kidneys under 20 Gy and 30 Gy doses [32]. In another report on the renal late effects treated for cancer in childhood, it has been shown that a radiation dose ≥10 Gy during photon irradiation is recognized as a risk factor for renal insufficiency; treatment more than ≥15 Gy is a high-risk factor for nephrotoxicity and renal dysfunction [17]. In our study, almost all pediatric patients received more than 20 Gy (RBE) of proton radiation to the kidneys, and over half of the children received more than 30 Gy (RBE) of radiation to the kidneys. However, during the long-term follow-up, there were no cases of renal dysfunction or deterioration of kidney function. These results may indicate that because the superior dose distribution of proton therapy enables only a portion of the kidney volume to be irradiated, the kidneys of children are spared to some extent and the effect on renal function become minimized. Although no more relevant evaluations of pediatric renal volume after photon therapy were found, this may be a new direction to explore in the future, which will be valuable in the comparison of growth of pediatric patients treated with different modalities.

Numerous studies have investigated renal effects after radiotherapy, but only a few have been conducted after PBT. In pediatric oncology, PBT has an outstanding therapeutic history, as shown by its ability to lower the risk of endocrine dysfunction, fertility decline, and second cancers in children and adolescents [9,33,34]. In Japan, PBT has been included in health insurance since 2016 as a treatment for pediatric tumors [35]. However, the late effects of PBT on different organs have not been sufficiently studied, especially for the kidneys in children and adults in long-term follow-up. In our previous study, findings have shown that unilateral renal atrophy after PBT was stronger in pediatric patients aged 4–7 years than in children aged 2–3 years [36]. This further promotes our intention to explore the comparison of renal late effects after PBT at different ages, such as pediatric and adult. Therefore, this study focused on changes in kidney volume and function after PBT and different responses were found in different age groups, with a significant difference in kidney volume changes in children and adults. Children had marked renal atrophy in follow-up, with a substantial decrease in the volume of the irradiated kidneys. The greater kidney atrophy in children may occur because organs and tissues are still growing and developing, making children more sensitive to radiation [37,38]. Children also have a higher relative risk of developing leukemia, brain, breast, skin, and thyroid cancers after exposure to radiation [39], in part due to their longer life expectancy giving a greater window of opportunity for manifestation of radiation damage [39,40]. Additionally, even in the present study, since the protons passed through the kidneys almost entirely, the probability is high that the post-Bragg peak distance would not have been a significant factor in the difference in renal atrophy between the two groups of patients. However, we must realize that children have smaller bodies than adults and less fat and other tissues, which may lead to different organ spacing and further effects the distribution of the dose. 

The impact of PBT on the volume of the irradiated kidneys in adults was minimal. This may be because adults possess a greater capacity for recovery from radiation injury. The regenerative ability of the renal parenchyma in adults may mitigate the effects of radiation-induced injury, leading to limited structural changes [41,42]. Additionally, the kidneys in adults are mature and stable, further reducing the likelihood of substantial changes after irradiation [43]. The slight atrophy of the control kidneys in adults can be attributed to several factors. First, as individuals age, organs undergo a certain degree of atrophy, which is a natural part of the aging process [44]. Additionally, the median follow-up period for adults was 70.8 months, and even minor changes in organ size can become more pronounced over time. Furthermore, although the control kidneys were not directly exposed to radiation, they might have had indirect exposure, leading to subtle atrophy. Different health conditions and lifestyles could also contribute to these outcomes [45]. These results indicate that age-related factors need to be considered when assessing late effects on the kidney after PBT, and children may be more susceptible than adults for these late effects. PBT remains a viable treatment option for adult patients with a favorable safety profile for the kidneys.

Although our study did not specifically analyze outcomes by gender factor, the potential for gender-related differences of PBT for kidney effects should be carefully considered. Previous research has indicated that hormonal and metabolic variations between genders could influence tissue effects to radiation, possibly affecting therapeutic outcomes and toxicity profiles [46]. Additionally, research has indicated that different cellular subtypes of pediatric tumors have distinct metabolic characteristics, therefore the response to the same treatment may vary [47]. Thus, future studies designed with the capacity should be explored this aspect in greater depth. Such studies should ideally include a larger cohort, allowing for a more stratified analysis that could offer clearer insights into the sex-specific responses to radiation, contributing to the optimization of personalized treatment strategies.

The significant negative correlation found between the dose-volume histograms (V10 and V20) and the relative kidney volume change per year suggest that V10 and V20 may serve as predictors of the degree of renal atrophy after PBT, particularly in pediatric patients. This information could aid in individualizing adjustments in the treatment plan and minimizing late renal effects. The assessment of eGFR and the absence of severe renal dysfunction in both age groups before and after irradiation indicated that overall renal function remained within acceptable limits, despite the observable and subtle changes in renal volume. This further confirms the safety of PBT with regard to renal health. The influence of surgery and chemotherapy on renal late effects should also be considered, since surgery has the potential to affect renal blood flow, angiogenesis, and overall renal function [48,49]. Renal function after abdominal radiotherapy in children may also be affected by concurrent chemotherapy, since most pediatric tumors are treated with chemotherapy [50], and chemotherapeutic agents may induce nephrotoxic effects, leading to kidney injury and renal impairment [51,52]. However, presently, we are unable to distinguish among renal impairments induced by surgery, chemotherapy, or radiotherapy. Future studies are needed to examine the combined effects of these treatment modalities to minimize renal complications. For adult males, benign prostatic hyperplasia (BPH) is a common condition that may lead to ureteral obstruction and subsequent kidney dysfunction and may increase the risk of radiation-associated damage. Even a slight amount of urinary obstruction due to prostate enlargement in male adults has been associated with kidney function deterioration [53]. In our study, only one adult male was found to have signs of BPH, but this did not further lead to ureteral obstruction or a reduction in kidney function. As the primary lesions in most patients in this study were located in the abdomen, pelvic imaging was rarely seen in subsequent follow-up observations, so the prostate volume and condition could not be well evaluated for these patients. However, we can confirm that none of the patients showed signs of ureteral obstruction, and there were no cases of kidney failure.

This study has certain limitations, including a small sample size and a short follow-up duration. Studies with larger sample sizes, extended follow-up periods, and more thorough exploration of other factors will augment understanding of late renal effects after PBT.

## 5. Conclusions

This study showed age-specific differences in renal late effects between children and adults after PBT. Irradiated kidneys in children exhibited significant atrophy, while adult kidneys were minimally affected by PBT. V10 or V20 might be predictors of the degree of renal atrophy after PBT, especially in children. The findings support the use of PBT as a promising therapeutic modality with minimal impact on renal function in both age groups.

## Figures and Tables

**Figure 1 cancers-16-01634-f001:**
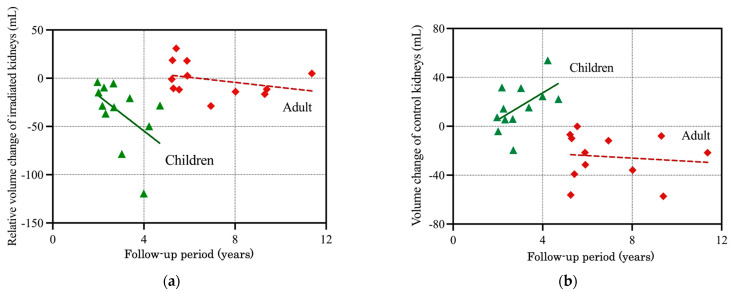
Volume changes in (**a**) irradiated kidneys relative to control kidneys and (**b**) control kidneys in pediatric (solid green line) and adult (dotted red line) patients. Atrophy of the irradiated kidneys was significantly greater in pediatric patients than in adults. The control kidneys of adults had mild atrophy after PBT, whereas the kidney size in children increased significantly due to normal growth and development.

**Figure 2 cancers-16-01634-f002:**
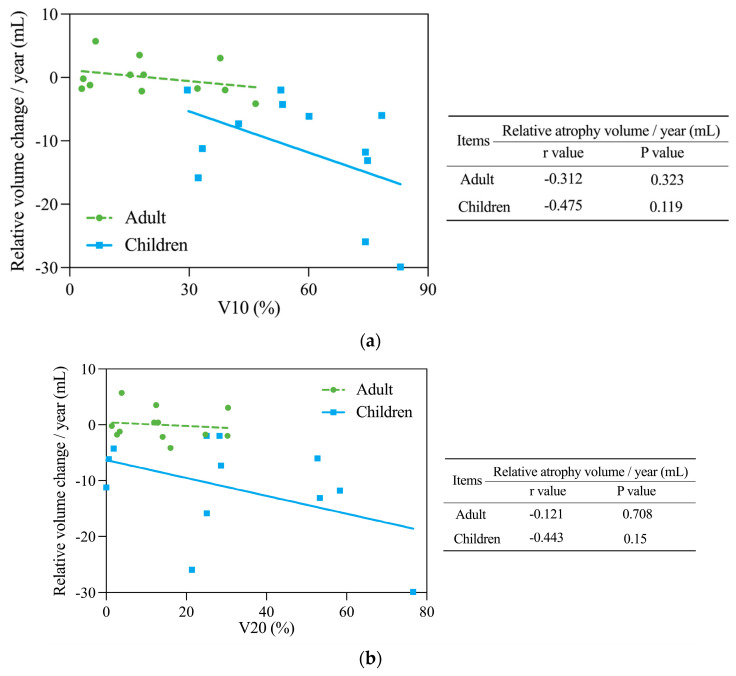
Correlation analyses for (**a**) V10 and (**b**) V20 with relative volume changes in irradiated kidneys/year in pediatric (solid blue line) and adult (dotted green line) patients. Pearson correlation coefficients (r) and significance (*p* value) are summarized on the right of each figure. V10 had a negative correlation with relative volume change/year in pediatric patients. This relationship was less pronounced in adults. V20 also had a negative correlation with relative volume change/year in pediatric patients, but almost no correlation in adults.

**Table 1 cancers-16-01634-t001:** Characteristics of patients, tumors and PBT.

Characteristics	Children (*n* = 12)	Adult (*n* = 12)
Age (years) Median Range	3.5 1–14	66 51–80
Sex Male Female	6 (50%) 6 (50%)	6 (50%) 6 (50%)
Primary disease HCC Neuroblastoma Osteosarcoma Ewing sarcoma Pancreatic cancer Rhabdomyosarcoma Metastatic liver cancer	- 8 (66.7%) 1 (8.3%) 1 (8.3%) - 2 (16.7%) -	6 (50%) - - - 2 (16.7%) - 4 (33.3%)
Surgery Gross total resection Subtotal resection Biopsy None	9 (75%) 1 (8.3%) 2 (16.7%) -	- 1 (8.3%) - 11 (91.7%)
Chemotherapy Pre-radiation Chemoradiation None	7 (58.3%) 5 (41.7%) 0 (0%)	1 (8.3%) 5 (41.7%) 6 (50%)
PBT Dose (range) (Gy (RBE)) Fractions (range)	30.6 (19.8–70.4) 17 (11–32)	72.6 (66–74) 22 (10–50)
Follow-up (months) Median Range	32.1 23.5–56.5	70.8 62.7–136.5

Data are shown as number (%) or as indicated in the table.

**Table 2 cancers-16-01634-t002:** Characteristics of treatment and changes in kidney volume after PBT in pediatric cases.

Case	Primary Disease	Age (y)	Total Dose (Gy (RBE))	Fractions	V10 ^a^ (%)	V20 ^a^ (%)	V30 ^a^ (%)	Follow-Up (Months)	Volume Change of Irradiated Kidney (mL) ^b^	Volume Change of Control Kidney (mL) ^b^	Relative Volume Change of Irradiated Kidney (mL) ^b,c^
#1	Neuroblastoma	10	30.6	17	53.04	28.26	0.06	23.5	+3.52	+7.39	−3.87
#2	Neuroblastoma	4	30.6	17	74.83	53.32	0	26.1	+3.27	+31.81	−28.54
#3	Neuroblastoma	2	30.6	17	78.4	52.74	15.3	56.5	−5.92	+22.29	−28.21
#4	Osteosarcoma	7	70.4	32	32.3	25.12	15.33	27.8	−31.08	+5.6	−36.68
#5	Rhabdomyosarcoma	1	50.4	28	29.55	25.07	21.53	32.0	+0.67	+5.96	−5.29
#6	Neuroblastoma	3	30.6	17	74.28	58.3	22.94	50.7	+4.13	+53.89	−49.76
#7	Neuroblastoma	3	19.8	11	60.14	0.66	0	40.6	−5.24	+15.44	−20.68
#8	Ewing sarcoma	14	55.8	31	83.06	76.58	68.74	47.9	−94.95	+24.5	−119.45
#9	Neuroblastoma	4	19.8	11	74.31	21.37	0	36.4	−47.29	+31.26	−78.55
#10	Neuroblastoma	3	19.8	11	53.45	1.85	0	27.0	+4.79	+14.4	−9.61
#11	Rhabdomyosarcoma	6	41.4	23	42.4	28.66	13.66	24.1	−18.74	−4.03	−14.71
#12	Neuroblastoma	3	30.6	17	33.31	0	0	32.2	−49.51	−19.45	−30.06

^a^ V10, V20, and V30 (%): volume irradiated by 10, 20, and 30 Gy (RBE) of PBT; ^b^ +: Volume increase; −: Volume reduction.; ^c^ Change of irradiated kidney volume compared to control kidneys.

**Table 3 cancers-16-01634-t003:** Characteristics of treatment and changes in kidney volume after PBT in adult cases.

Case	Primary Disease	Age (y)	Total Dose (Gy (RBE))	Fractions	V10 ^a^ (%)	V20 ^a^ (%)	V30 ^a^ (%)	Follow-Up (Months)	Volume Change of Irradiated Kidney (ml) ^b^	Volume Change of Control Kidney (ml) ^b^	Relative Volume Change of Irradiated Kidney (mL) ^b,c^
#1	HCC	59	72.6	22	15.17	11.94	9.79	136.5	−16.74	−21.61	+4.87
#2	Metastatic liver cancer	80	72.6	22	3.01	2.68	2.49	111.7	−24.36	−7.92	−16.44
#3	Metastatic liver cancer	69	72.6	22	3.42	1.39	0.6	62.7	−7.97	−6.87	−1.10
#4	Metastatic liver cancer	51	72.6	22	5.11	3.38	3.07	112.7	−68.7	−57.26	−11.44
#5	HCC	78	72.6	22	18.1	14.1	11.36	66.6	−12.02	−0.16	−11.86
#6	Metastatic liver cancer	58	72.6	22	6.5	3.86	2.19	65.0	−8.23	−39.15	+30.92
#7	HCC	64	72.6	22	32.08	24.75	19.67	96.3	−49.91	−35.86	−14.05
#8	HCC	78	74	37	18.57	12.94	6.37	70.9	−28.93	−31.47	+2.54
#9	Pancreatic cancer	52	67.5	50	46.69	16.04	1.6	83.3	−40.58	−11.73	−28.85
#10	Pancreatic cancer	61	67.5	50	17.55	12.45	4.83	63.1	−37.6	−56.18	+18.58
#11	HCC	79	72.6	22	37.84	30.4	23.32	70.7	−3.48	−21.51	+18.03
#12	HCC	68	72.6	22	39.06	30.29	23.74	63.6	−20.49	−9.93	−10.56

^a^ V10, V20, and V30 (%): volume irradiated by 10, 20, and 30 Gy (RBE) of PBT; ^b^ +: Volume increase; −: Volume reduction.; ^c^ Change of irradiated kidney volume compared to control kidneys.

**Table 4 cancers-16-01634-t004:** Correlation coefficients and significance for relationships of V10 and V20 with the relative volume change in irradiated kidneys in all patients.

Items	Relative Volume Change/Year (mL)	
	r Value	*p* Value
V10	−0.701	<0.001
V20	−0.559	0.005

**Table 5 cancers-16-01634-t005:** Number of cases in different CKD stages evaluated with eGFR before, during, and after PBT.

	Adults (*n* = 12)			Children (*n* = 12)		
CKD Stage	Before PBT	During PBT	After PBT	Before PBT	During PBT	After PBT
**1**	5	3	2	11	10	10
**2**	6	8	8	1	2	2
**3**	1	1	2	-	-	-

## Data Availability

Data are available from the corresponding author upon request.

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
