# Peer review of "Late Changes in Renal Volume and Function after Proton Beam Therapy in Pediatric and Adult Patients: Children Show Significant Renal Atrophy but Deterioration of Renal Function Is Minimal in the Long-Term in Both Groups"

_cancers, 2024, doi:10.3390/cancers16091634_

Round 1
Reviewer 1 Report
Comments and Suggestions for Authors
I am grateful for the opportunity to review manuscript # cancers-2919851, entitled “Differential late effects on the kidneys after proton beam therapy in pediatric and adult patients.”
The authors report follow-up of 12 pediatric (6 male and 6 female) and 12 adult (6 male and 6 female) who received abdominal PBT, each involving irradiation of the ipsilateral kidney. In general, the authors’ finding suggest that pediatric patients suffered significant atrophy of the ipsilateral kidney, whereas the adults did not. V10 and V20 were markers for atrophy, allowing for tailoring planning. Both age groups experienced virtually no reduction in kidney function. Perhaps a more significant finding was that the contralateral control kidney in pediatrics continued to grow post-PBT, and so possibly enabling compensation for the ipsilateral atrophy.
The manuscript is clear and concise. This is a nice and useful study, albeit with a small cohort.
I only have one comment before recommending publication:
It would be useful to compare the degree of pediatric kidney atrophy seen here with any photon-based radiotherapy literature. If the levels are atrophy are similar between the modalities, but the contralateral kidneys are spared better with PBT and allowed to grow better, than that provides additional evidence in favor of PBT for pediatrics in this context. A paragraph comparing and discussing this would be a good addition.
Author Response
Thank you for taking the time to review our manuscript and your kind comments. We really appreciate your recognition of our work. Please see the attachment and the corresponding corrections highlighted in the re-submitted files.

Reviewer 2 Report
Comments and Suggestions for Authors
The manuscript by Li et al. “Differential Late Effects on the Kidneys after Proton Beam Therapy in Pediatric and Adult Patients” reports proton radiation toxicity on kidneys of young and old patients through measurement of kidney size and function. It appears that proton radiation toxicity is more pronounced in pediatric patients than in adults. Authors provided a generic explanation for their findings without discussing potential source of difference between pediatric and adult patients in terms of body size, thus post-Bragg peak distance between the primary radiation target and incidental organs. They also did not specify the exact path for radiation for small groups of patients, and a comparison with photon radiation might be more informative in examining the underlying differential late effects on kidneys. Overall this study is weak for its significance both clinically and scientifically, thus is suitable for a more specialized journal.
Comments on the Quality of English Language
Technical English writing is clear.
Author Response
Thank you for taking the time to review our manuscript and your kind comments. We really appreciate your contribution to our work. Please see the attachment and the corresponding corrections highlighted in the re-submitted files.

Reviewer 3 Report
Comments and Suggestions for Authors
This manuscript is interesting and offers some new data and novel findings in the field of kidney injury related to radiotherapy on adjacent tumors, both in pediatric and adult patients, of both sexes. Despite the small number, the study group is well balanced, 12 pediatric patients (6 males and 6 females) and 12 adult ones (6 males and 6 females).
This manuscript is well written, and requires only a number of minor changes before being accepted for publication in Cancers. The changes needed are detailed in the text below. These changes could significantly improve the manuscript's originality and clinical significance.
TITLE: The most important finding of this paper, which is also its main strength, is that the long-term impact of PBT is different in pediatric and adult patients, but there is, as a late effect, a minimal impact on renal function in both age groups. This is the real original finding.
This is a proposed change for the TITLE: which puts more attention on the study final findings:
LATE CHANGES IN RENAL VOLUME AND FUNCTION AFTER PROTON BEAM THERAPY IN PEDIATRIC AND ADULT PATIENTS: CHILDREN SHOW SIGNIFICANT RENAL ATROPHY, BUT IN BOTH AGE GROUPS DETERIORATION OF RENAL FUNCTION IS MINIMAL IN THE LONG-TERM
ABSTRACT AND SIMPLE SUMMARY: V10 and V20 should not be mentioned in the abstracta and simple summary, without an explanation. It would be best to use V10 and V20 for the first time in the main text, and explain for they mean.
MATERIALS AND METHODS:
General comment: Why was the median follow up so different in pediatric (32 months) and adult patients (70 months)? The Authors should explain this point
RESULTS The response of the irradiated kidneys after PBT clearly differed in adult and pediatric patients
1. The control kidney became larger is children as they grew, whereas those of adults showed slight atrophy
2. There are clear differences in the response of control and irradiated kidneys after PBT
3. The relative volume change of the kidneys after irradiation in all patients showed the strongest significant correlation with V10 and was also significantly correlated with V20.
It would be very interesting if the Authors would report any different in gender-related radiation-associated kidney damage. This is a difficult but highly clinically significant task.
Did the females show any difference as compared to the males? It has been shown, in recent publications, that there can be very interesting gender-associated differences in kidney function and response to treatment, either is oncological (Mancini M., et al, Gender-related approach to kidney cancer management: moving forward. International Journal of Molecular Sciences 21.9: 3378; 2020) or in functional clinical situations (Ishiyama Y., et al., Association between ureteral clamping time and acute kidney injury during robot-assisted radical cystectomy. Curr Oncol 2021,28:4986-4997: doi:10.339/curroncol28060418).
Since the study group is so well balanced in terms of gender (six males and six females in both group), the Authors should make an effort to at least mention the possibility of gender-related difference response to radiation damage in the reported cases.
DISCUSSION:
General comment: The discussion section is the place where the results are compared to what has been published before. Here you must stress the strength and the originality of your work.
The presence of urinary obstruction due to prostate enlargement in adult males could itself lead to increased kidney radiation-associated damage, and this should be evident only in adults, not in pediatric patients. It would be clinically relevant if the Authors could mention this point in the Discussion, since this is another very relevant clinical point. Even slight amount of urinary obstruction due to prostate enlargement in male adults, has been associated to kidney function deterioration (Righetto M., et al: Patients with renal transplants and moderate to severe LUTS benefit from urodynamic evaluation and early transurethral resection of the prostate. World J Urol 2021,39:4397-4404. Doi:10.1007/s00345-021-03799- This concept should be mentioned by the Authors in the Discussion. It also would be worth to report in the Results prostate volume or presence/absence of obstruction in the adult males reported in the study.
Author Response

(The authors gave the same response as above.)

Round 2
Reviewer 1 Report
Comments and Suggestions for Authors
Authors have addressed reviewer comments adequately with this revision.
Reviewer 2 Report
Comments and Suggestions for Authors
Authors modified the manuscript accordingly.